# Bimodal Imaging of Tumors via Genetically Engineered *Escherichia coli*

**DOI:** 10.3390/pharmaceutics14091804

**Published:** 2022-08-27

**Authors:** Linlin Zhang, Yuanyuan Wang, Dengjin Li, Liang Wang, Zhenzhou Li, Fei Yan

**Affiliations:** 1Department of Ultrasound, The Second People’s Hospital of Shenzhen, The First Affiliated Hospital of Shenzhen University, Shenzhen 518061, China; 2Shantou University Medical College, Shantou 515041, China; 3Center for Cell and Gene Circuit Design, CAS Key Laboratory of Quantitative Engineering Biology, Shenzhen Institute of Synthetic Biology, Shenzhen Institutes of Advanced Technology, Chinese Academy of Sciences, Shenzhen 518055, China; 4Center for Quantitative Synthetic Biology, CAS Key Laboratory of Quantitative Engineering Biology, Shenzhen Institute of Synthetic Biology, Shenzhen Institutes of Advanced Technology, Chinese Academy of Sciences, Shenzhen 518055, China; 5Research Laboratory for Biomedical Optics and Molecular Imaging, CAS Key Laboratory of Health Informatics, Shenzhen Institute of Advanced Technology, Chinese Academy of Sciences, Shenzhen 518055, China

**Keywords:** *Escherichia coli*, bimodal imaging, acoustic imaging, optical imaging, bioluminescence, plasmid construction

## Abstract

Although there are emerging innovations of molecular imaging probes to detect and image tumors, most of these molecular dyes and nanoparticles have limitations of low targetability in tumors and fast clearance when administered systemically. In contrast, some bacteria, such as *Escherichia coli MG1655*, can selectively proliferate in a hypoxic environment inside of a tumor for several days, which highlights the potential for the development of a genetically encoded multimodal imaging probe to monitor the progress of the tumor. Here, we developed bimodal imaging tumor-homing bacteria (GVs-miRFP680 MG1655) that allow both optical and acoustic imaging in tumor-bearing mice. An in vivo optical image system and a Vevo 2100 imaging system were applied to detect different imaging properties of the engineered bacteria in vivo. Our results show that the GVs-miRFP680 MG1655 bacteria can effectively integrate the advantages of low tissue absorbance from near-infrared fluorescent proteins and non-invasiveness from gas vesicles. We successfully developed GVs-miRFP680 MG1655 bacteria, which have both acoustic and optical imaging abilities in vitro and in vivo. The acoustic signal can last for up to 25 min, while the near-infrared fluorescence signal can last for up to 96 h. The combination of different imaging modalities in the tumor-homing bacteria may contribute to the non-invasive monitoring of the therapeutic effect of bacterial therapy in the future.

## 1. Introduction

Over the past decade, bacteria therapy has converged interests for its noticeable priority in genetic engineering and selective tumor targetability. Many anaerobic and facultatively anaerobic bacteria, such as *Salmonella*, *Clostridium,* and *Escherichia coli* (*E. coli*), are genetically engineered to act as a therapeutic drug delivery system or coordinate with chemotherapy and immunotherapy to improve their therapeutic effects [1,2,3,4,5,6]. Acting as a microbe delivery platform, bacteria can actively home to tumor sites and continue to proliferate inside of a tumor, achieving a longer time of retention [7,8]. Furthermore, bacteria can be conveniently reprogrammed via transformation with plasmids, which combine different genetic information into one microbe to enhance its tumor-homing ability or improve the anti-tumor efficacy [3,9,10]. As for tumor treatments, it is very important to monitor their concentration and confirm their localization in the tumor, especially for living bacterial carriers. To date, many delivery systems should be endowed with optical or acoustic imaging properties to visualize their distribution in live animals. However, most of the optical (e.g., nanoparticles, dyes) and acoustic (e.g., gas vesicles) imaging probes have limitations of fast clearance and low specificity in the tumors [11]. In the near future, a non-invasive microbe imaging probe may be achieved by genetically engineering the bacteria with different reporter genes to enable the visualization of them in living animals.

The acoustic reporter gene (ARG) is a genetic construct that encodes gas vesicles (GVs) which enables bacteria to be visualized in vivo using a high-frequency diagnostic ultrasound probe [12]. GVs are a class of gas-filled nano-scaled bubbles that were first found in photosynthetic bacteria and archaea in nature. The nanobubbles are composed of amphiphilic protein shells permeable to gas. The acoustic imaging property of GVs was already demonstrated in heterogeneously expressed bacteria [12,13,14]. However, the combination of GVs with other imaging modalities in the bacteria to achieve multimodal non-invasive imaging in vivo has not been investigated yet.

Bioluminescent proteins and visible-light fluorescent proteins (FPs) are widely used as optical imaging reporters for cell or bacterium tracking in vivo. In recent years, near-infrared fluorescent proteins (NIRFPs) have gradually overridden visible-light FPs in the application of in vivo imaging by the priority in optical windows (650–900 nm). In this spectrum wavelength range, NIRFPs exhibit the characteristics of low tissue absorbance, low phototoxicity, high effective brightness, and deep tissue penetration [15,16,17]. After several rounds of mutation and selection of variants, miRFP680, one kind of monomerized product from NIRFPs first published in 2020, demonstrated the highest quantum yield and molecular brightness among all of the miRFP variants in mammalian cells [15]. This exhibits its promising application as a new generation of non-invasive NIR probes in living animals [18,19].

Here, we report the development of a bimodal microbe imaging probe (GVs-miRFP680 MG1655) by successfully combining acoustic GVs and optical miRFP680 into *Escherichia coli* MG1655, which has been widely used in bacterial tumor therapy [2,20]. Using a high-frequency diagnostic ultrasound probe, the acoustic imaging property of bacteria could be detected after intratumoral injection, and the signal intensity continued to maintain for up to 25 min. The systemic injection of GVs-miRFP680 MG1655 shows selective bacterial colonization in the tumors via optical imaging devices. As the bacteria can proliferate and accumulate in the tumor sites, the engineered bacteria provided stable optical signals over several days. A bimodal microbe imaging probe would be a valuable attempt in the development of a bacteria-based therapeutic platform since it provides more information on the different imaging modalities. Moreover, facilitating the visualization of bacteria inside the tumors by both optical and acoustic imaging can help promote future progress in a more concise evaluation of the therapeutic effect of bacteriotherapy.

## 2. Materials and Methods

### 2.1. Bacterial Strains and Plasmid Construction

The wild-type *E. coli* K-12 strain (MG1655, ATCC 700926) used in this study was purchased from the American Type Culture Collection (ATCC). The commercially available plasmid pTD103-ARG1 (plasmid #106475) was purchased from Addgene. In order to ensure plasmid compatibility, the DT012 expression plasmid (obtained from Chenli Liu’s lab at the Shenzhen Institute of Advanced Technology, Chinese Academy of Sciences (SIAT)) was constructed via Gibson assembly, which contains a p15A origin and chloramphenicol resistance, was used as the starting plasmid, and was fused using a PCR-based method to eliminate the original promoter and create a DT012 vector backbone. Then, the miRFP680 gene fragment and HO fragment with a revised constitutive T7 promoter were fused using the PCR-based method from the original plasmid pJC-miRFP680-HO (obtained from Jun Chu’s lab at SIAT). The original plasmid was constructed via Gibson assembly. The above fragments were inserted into the DT012 vector via Gibson assembly to create the constitutively expressed plasmid DT012-miRFP680.

### 2.2. Transformation and Construction of GVs-miRFP680 MG1655

The plasmid pTD103-ARG1 was first transformed into a wild-type *E. coli* K-12 strain (MG1655, ATCC 700926) to select stable clones (GVs-MG1655) expressing GVs. Then, the selected GVs-MG1655 strain was used for preparing electrocompetent cells. The plasmid DT012-miRFP680 was electroporated into the GVs-MG1655 electrocompetent cells to create GVs-miRFP680 MG1655. The clone emitting high fluorescence was selected using a Caliper in vivo imaging system (IVIS) spectrum and a Tecan infinite M1000 pro microplate reader.

### 2.3. Bacterial Culture and Expression of GVs-miRFP680 MG1655

The bacterial culture follows the previously published protocol [12], except the bacteria were cultured in LB broth with 25 µg/mL of kanamycin and 25 µg/mL of chloramphenicol. For the expression of plasmid pTD103-ARG1, the cells were cultured to OD600 = 0.5 and then induced with 3 nM of N-(β-ketocaproyl)-L-homoserine lactone (AHL) for 24 h at 30 °C. DT012-miRFP680 was constitutively expressed, and no induction was required.

### 2.4. Characteristics of Gas Vesicles in GVs-miRFP680 MG1655

The GVs-miRFP680 MG1655 was observed by transmission electron microscopy (TEM, Hitachi H-7650 Tokyo, Japan) and phase-contrast microscopy (Olympus IX83 inverted microscope, Tokyo, Japan) following the appropriate protocols [12,14].

### 2.5. Cell Culture

The murine bladder cancer cell line, MB49, was purchased from ATCC (Manassas, VA, USA). MB49 cells were cultured in RPMI 1640 medium (Corning, New York, NY, USA) that contained 10% fetal bovine serum (Gibco, 10099141C, New York, NY, USA) and incubated at 37 °C in a 5% CO_2_ humidified incubator.

### 2.6. Animal Models

All of the experimental animal procedures were approved by the management group of experimental animal nursing institutions of the Technical Research Institute of the Shenzhen Institute of Advanced Technology, Chinese Academy of Sciences (approval number: SIAT-IACUC-210303-HCS-YF-A1693; date of approval: 10 September 2021). Briefly, six-week-old female C57 mice were purchased from the Charles River Laboratory (CRL), Beijing. To establish the tumor models, 5 × 10^6^ MB49 mouse bladder carcinoma cells were injected subcutaneously into the right flank of the mice. The tumor volume (mm^3^) was calculated using the formula (length × width × height) × 0.5 in millimeters. The size of the tumors was monitored and permitted to grow to a size of 6–10 mm in diameter over two weeks. None of these tumors reached the maximal tumor size of 1500 mm^3^. After the tumor volume reached an average size of 120 mm^3^, all of the experimental animals were randomly assigned to different groups in the following experiments.

### 2.7. Ultrasound Imaging Capability of GVs-miRFP680 MG1655

The in vitro ultrasound imaging capability of GVs-miRFP680 MG1655 was demonstrated through gel phantoms according to the previously published method [12]. When the tumor volume (in the right flank) reached an average size of 120 mm^3^, mice were randomly divided into four groups. The imaging probe (21 MHz probe, 50% power) was settled 0.5 cm above the center of the mice tumor, with the surface filled with couplant to enable the transmission of the acoustic beam. Then, different concentrations of bacteria cells (0, 0.5 × 10^9^, 1.0 × 10^9^, and 2.0 × 10^9^ colony-forming unit (CFU)/mL) in 100 μL of PBS was injected into the tumor. After that, B-mode and contrast-mode images were acquired by the VisualSonics Vevo-2100 ultrasound imaging system. In vivo ultrasound images were pseudo-colored, as indicated in the corresponding color scale bar. The mean signal intensity was calculated using Image J.

### 2.8. Optical Imaging Capability of GVs-miRFP680 MG1655

The in vitro optical imaging capability was evaluated using the Tecan infinite M1000 pro microplate reader and normalized by the optical density of the bacteria according to the previous report [21]. Confocal microscopy was used to visualize the fluorescence of a single bacterium. When the tumor volume (in the right flank) reached an average size of 120 mm^3^, mice were randomly divided into two groups. Then, 5 × 10^7^ CFU GVs MG1655 or GVs-miRFP680 MG1655 in 100 μL of PBS was injected via the tail vein. After that, fluorescence images were acquired at 0, 24, 48, 72, 96, and 120 h after injection using IVIS Lumina II (Caliper Life Sciences, Hopkinton, MA, USA; excitation filter: 640 nm, emission filter: 680 nm). Three mice were sacrificed by euthanasia at the time of highest fluorescence accumulation in the tumors after bacterial injection; the tumors and normal organs (heart, liver, spleen, lung, and kidney) were collected for the acquisition of fluorescent signal intensity (IVIS Lumina II Caliper Life Sciences, USA; excitation filter: 640 nm, emission filter: 700 nm). The tumors were used for tissue sectioning, and the slices were stained with DAPI (blue) and the anti-CD31 antibody (green) for confocal imaging according to the standard method.

### 2.9. Statistical Analysis

Data are presented as mean ± s.d. Statistical analysis was performed using GraphPad Prism 7 (GraphPad Software Inc., San Diego, CA, USA). The means of different groups were compared using the one-way ANOVA test. Differences showing a *p*-value < 0.05 were considered statistically significant.

## 3. Results

### 3.1. Construction of GVs-miRFP680 MG1655 Bacteria

The wild-type MG1655 was used to construct the GVs-miRFP680 MG1655 bacteria. Briefly, plasmid pTD103-ARG1 (Figure 1b) was electroporated into the wild-type MG1655 to obtain the GVs-MG1655 bacteria. Considering that the origin of replication of the pTD103-ARG1 plasmid is similar to the original plasmid of miRFP680 (plasmid pJC- miRFP680-HO), in this study, we constructed the plasmid DT012-miRFP680 (Figure 1c) by inserting the cassette containing the miRFP680 gene and the heme oxygenase (HO) gene under a modified T7 promoter into a biocompatible p15A origin skeleton. The cassette containing the heme oxygenase (HO) would produce the linear tetrapyrrole biliverdin IXα (BV) as a chromophore for miRFP680. A modified T7 promoter was used to express the exogenous genes without IPTG induction. The DT012-miRFP680 plasmid was transformed into the GVs-MG1655 competent bacteria cells to obtain the target GVs-miRFP680 MG1655 bacteria (Figure 1a).

### 3.2. In Vitro Ultrasound Imaging of GVs-miRFP680 MG1655

The previous evidence showed that the GV gene cluster could be stably expressed in heterogenous bacteria, such as *Escherichia coli* [12,13,14]. From the phase-contrast microscopy and transmission electron microscope, numerous GVs can be clearly observed in the engineered GVs-miRFP680 bacteria (Figure 2), while miRFP680 MG1655 bacteria show no visible GVs. In order to determine the most suitable inducer concentration for the engineered bacteria to produce GVs, we induced the engineered bacteria (OD_600_ = 0.5) with 10, 50, 100, 300, and 500 nM N-(β-ketocaproyl)-L-homoserine lactone (AHL). The B-mode and contrast mode clearly recorded the signals produced by GVs (Figure 3a). When 100 nM of AHL was used, the ultrasound signal of the GVs-miRFP680 MG1655 was the brightest among the other concentration groups (*p* < 0.0001) (Figure 3a–c). By contrast, the miRFP680-MG1655 group as the control could not produce any significant ultrasound imaging signal despite the above concentration of inducer was given (Figure 3a–c). According to the results acquired, we used 100 nM of AHL concentration to induce the expression of pTD103-ARG1 in subsequent experiments. To observe the changes in signal intensity with time, we applied the 100 nM inducer concentration to the GVs-miRFP680 MG1655 bacteria and the miRFP680-MG1655 bacteria. When the engineered bacteria were induced for more than 4 h, the contrast signal for the GVs-miRFP680 MG1655 bacteria became visible. The mean signal intensity of the engineered bacteria gradually increased and reached a peak value at 24 h (Figure 3d,e). Meanwhile, the level of signal intensity of the engineered bacteria had a positive correlation with the bacteria density (*p* < 0.001) (Figure 3f–h). By contrast, the miRFP680-MG1655 group did not detect any significant change in the ultrasound signal over time or with an increase in the density of bacteria, which denied the possibility of interference of the signal from a high concentration of bacteria (Figure 3d–h).

### 3.3. In Vivo Ultrasound Imaging of GVs-miRFP680 MG1655

To demonstrate the imaging capability in vivo, we next applied the engineered bacteria to the tumor-bearing mice. After a 24 h induction with 100 nM of AHL, GVs-miRFP680 MG1655 with different bacteria densities (0, 0.5 × 10^9^, 1.0 × 10^9^, and 2.0 × 10^9^ CFU/mL) were injected into the tumors. In vivo ultrasound images of the tumor were taken at different time points (Figure 4a). The results showed that 0.5 × 10^9^ CFU of cells could produce a detectable signal inside the tumor (Figure 4a). The ultrasound signal in each group was maintained for about 25 min. Notably, as the density of bacteria increased, the ultrasound signal became more and more intense (*p* < 0.001) (Figure 4a–c). This result was reasonable when considering the fact that the number of gas vesicles inside the engineered bacteria reflected more ultrasound signals, along with an increased number of bacteria in the tumor. The gradual decay of acoustic signals can be attributed to the hydrodynamics of fluid with the infiltration of bacteria into the surrounding interstitial fluid. Meanwhile, the injection of an equal volume of lysogeny broth (LB) culture medium did not produce any detectable signal (Figure 4a). As shown in Figure 4b,c, we can see that the peak intensity produced by 0, 0.5 × 10^9^, 1.0 × 10^9^, and 2.0 × 10^9^ CFU bacteria in the tumors were 2.297 ± 0.2739, 20.96 ± 3.701, 28 ± 3.72, and 48.56 ± 5.63 a.u., respectively. Combined with the data in vitro, we can further confirm that the engineered GVs-miRFP680 MG1655 bacteria can effectively produce relatively high acoustic signals in vivo due to the presence of GVs.

### 3.4. In Vitro Optical Imaging of GVs-miRFP680 MG1655

In order to visualize the optical imaging signal, we examined the engineered GVs-miRFP680 MG1655 bacteria using confocal microscopy. Since miRFP680 is characterized by the excitation wavelength maximum at 661 nm and emission wavelength maximum at 680 nm [15], we attributed the Cy5 channel (emission: 640 nm, excitation: 663–738 nm) to detect them. Both groups exhibited white round areas inside, which demonstrated the expression of GVs. Notably, except for the area occupied by GVs, the GVs-miRFP680 MG1655 bacteria were filled with pink pseudo-colored fluorescence, while the control group GVs-MG1655 bacteria could not emit any fluorescence in this range of wavelength (Figure 5a). Optical density (OD_600_)-normalized fluorescence as a function of culture time illustrated the quantitative difference between GVs-miRFP680 MG1655 and GVs MG1655 (Figure 5b). These data demonstrated the successful expression of both NIRFP fluorescence and GVs in GVs-miRFP680 MG1655 bacteria.

### 3.5. In Vivo Tracking Capability of the Tumor-Homing Characteristic of GVs-miRFP680 MG1655

To further demonstrate the tracking capability of GVs-miRFP680 MG1655 to home to tumor sites, we intravenously injected 5 × 10^7^ CFU of GVs-miRFP680 MG1655 or GVs MG1655 bacteria into the tumor-bearing mice. Then, the tumor-bearing mice were monitored every 24 h using an in vivo imaging system. As shown in Figure 6a, the engineered bacteria gradually aggregated in the tumor after 24 h and maintained a significant level of fluorescence for up to 96 h, with a peak fluorescence intensity acquired at 48 h. In contrast, the GVs-MG1655 group as a control did not produce a significant fluorescence signal during this process. Further, the ex vivo examination of the large organs collected at 48 h showed that the GVs-miRFP680 MG1655 bacteria significantly accumulated in the liver, kidney, and tumor, while organs in the GVs-MG1655 group did not exhibit obvious NIRFP fluorescence (Figure 6b,c). To further confirm this, the extracted tumors were removed, sliced, and stained with DAPI (blue) and the anti-CD31 antibody (green). Figure 6d demonstrated that the GVs-miRFP680 MG1655 group showed detectable fluorescence inside the tumor, whereas the GVs MG1655 control group showed no visible NIRFP fluorescence at each tumor section. Notably, pink pseudo-colored fluorescence could be observed inside the blood vessels (green, anti-CD31-antibody-stained). These data demonstrated the optical imaging tracking and tumor-homing capabilities of GVs-miRFP680 MG1655 bacteria in vivo.

## 4. Discussion and Conclusions

Gram-negative facultatively anaerobic bacteria such as *E. coli MG1655* are widely used as genetic vectors to express exogenous proteins for tumor bacteriotherapy. Due to the facultative anaerobic characteristics of this species, *E. coli MG1655* can effectively colonize and self-replicate in solid tumors [2,7,20]. Despite the increasingly prominent roles of bacteria in cancer therapy, few of them have been applied to clinical stages. There are still fundamental obstacles to clinical applications of bacterium-based tumor therapy. One of them lies in the difficulty of visualizing the microbes homing to tumor sites.

Ultrasound imaging is a non-invasive imaging modality that can produce real-time imaging information, with the advantages of excellent spatial resolution and high tissue penetration. These superiorities determine its promising application in visualizing the function of engineered cells non-invasively inside opaque organisms [13]. Previous researchers have engineered bacteria with the capacity to image with clinically used high-frequency ultrasound probes [12,22]. However, despite intratumoral imaging, the continuous monitoring of tumor-homing bacteria requires more systemic imaging information, which confirms that the engineered bacteria can still colonize inside the tumor after passing through the blood circulation. Previously, one of the researchers has already developed bacterial luciferase (Lux)-expressing bacteria with the real-time monitoring property of bacterial migration inside small living animals [8]. This work was achieved by the lambda Red system. Although fluorescence imaging alone was demonstrated to have the capacity to provide enough information about the distribution of the bacteria inside living animals, the supplementary information acquired from ultrasound imaging at the tumor sites can help to provide more qualitative information about the tumor. In order to design a bacterial imaging probe with both positioning value and diagnostic value, in this study, we genetically engineered the tumor-targeting bacteria by introducing the acoustic reporter gene and NIRFP gene into *E. coli MG1655*, providing it with a bimodal imaging capability for tracking inside the tumor. In order to integrate the above function, we chose to use expressive plasmids to build up this system. We used the acoustic reporter gene (ARG1) to express GVs, which can provide a backscatter property to enable ultrasound imaging of the engineered bacteria. We also used a new generation of fluorescent proteins: near-infrared fluorescent protein miRFP680, which demonstrated the highest quantum yield and molecular brightness among all of the miRFP variants in mammalian cells [15]. The combined expression of heme oxygenase provides miRFP680 with BV molecules as chromophores, without the demand for an extra exogenous supply of chromophores. Moreover, with the advantage of low tissue absorbance, miRFP680 can function well to accompany the genetically modified bacteria. The bimodal imaging capability of our engineered bacteria can help to monitor the colonization of therapeutic bacteria inside the solid tumor.

In this study, we successfully developed GVs-miRFP680 MG1655 bacteria, which have acoustic and optical imaging abilities both in vitro and in vivo. However, the shortcomings of this engineered system should also be addressed. Although we can achieve different imaging functions by constructing and co-expressing several functional plasmids inside the bacteria, the level of expression of the double-plasmid system is unstable. Theoretically, considering the long fragments of the target genes, genomic engineering with the lambda Red system combined with the plasmid expression may be a rational choice for future progress in this aspect. However, the expression level of genome editing is usually much lower. The trend of future research could also address more on the potential of bacterial vectors to achieve more therapeutic effects. The strategy of using nanoantibodies and immune-related molecules is gradually being applied in many research studies to enhance the capacity of the engineered bacteria against different types of cancer [1,3,14,23]. Facilitating the visualization of bacteria inside the tumors with both optical and acoustic imaging can help promote future progress in concisely managing the therapeutic effect of bacteriotherapy.

## Figures and Tables

**Figure 1 pharmaceutics-14-01804-f001:**
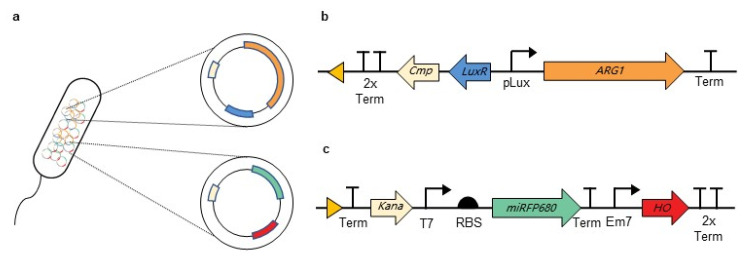
The construction of GVs-miRFP680 MG1655 bacteria. (**a**) Schematic illustration of the GVs-miRFP680 MG1655 bacteria. (**b**) pTD103-ARG1 plasmid map. ARG1 refers to the acoustic reporter gene 1. (**c**) DT012-miRFP680 plasmid map. miRFP680 refers to the monomeric near-infrared fluorescent protein 680. HO refers to heme oxygenase.

**Figure 2 pharmaceutics-14-01804-f002:**
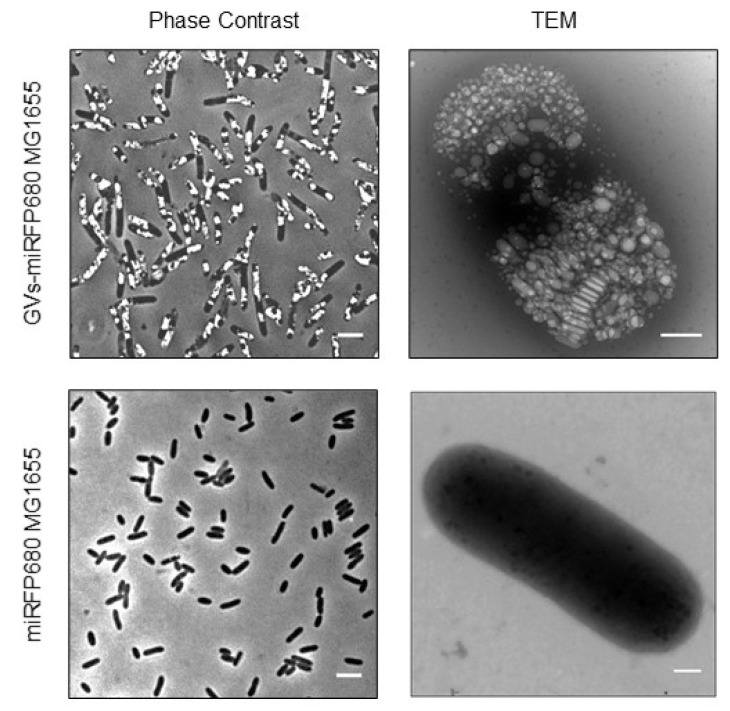
The characteristics of GVs-miRFP680 MG1655. The engineered bacteria GVs-miRFP680 MG1655 filled with GVs (**upper**) and the control group bacteria miRFP680 MG1655 (**lower**). Scale bars = 5 μm. Transmission electron microscope (TEM) images (**right**) depict the distribution of GVs inside GVs-miRFP680 MG1655 (**upper**) and miRFP680 MG1655 (**lower**). Scale bars = 200 nm.

**Figure 3 pharmaceutics-14-01804-f003:**
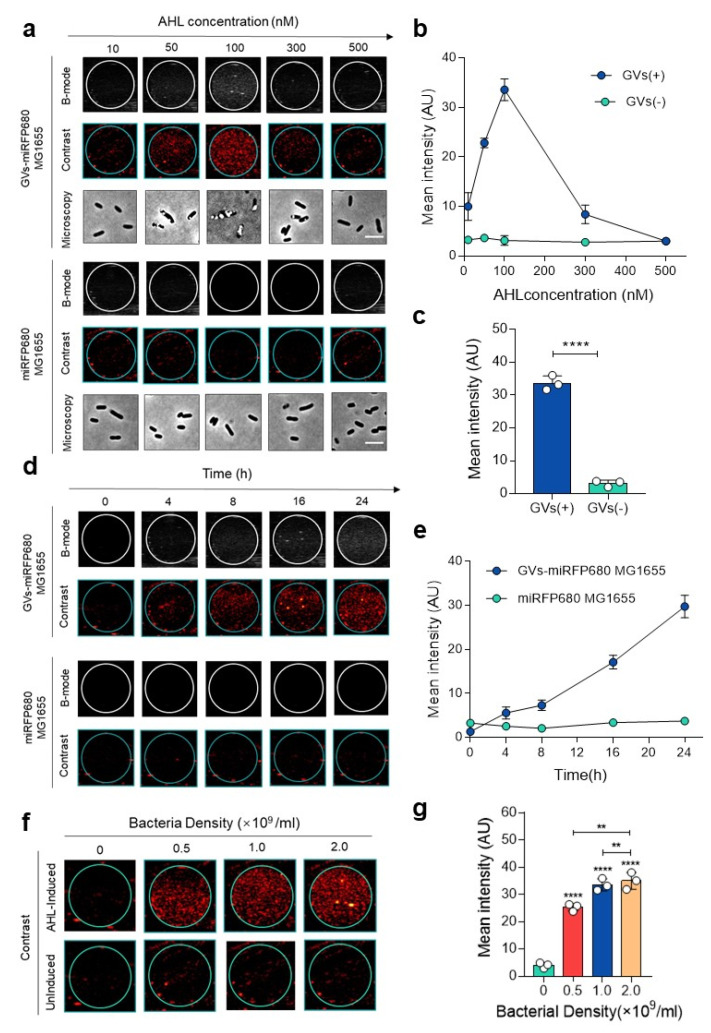
In vitro ultrasound imaging characteristics of GVs-miRFP680 MG1655. (**a**) Ultrasound images of GVs-miRFP680 MG1655 and miRFP680 MG1655 (control) induced by different concentrations of AHL for 24 h. Scale bars = 5 μm. (**b**) Quantitative analysis of the imaging signal intensities in (**a**). Data represent the mean ± SD from 3 independent experiments. (**c**) Mean signal intensity of bacteria induced by 100 nM AHL after 24 h. GVs(+) represents the GVs-miRFP680 MG1655 group in (**a**), GVs(−) represents the miRFP680 MG1655 group in (**a**). **** *p* < 0.0001 vs. control. (**d**) Ultrasound images of GVs-miRFP680 MG1655 and miRFP680 MG1655 (control) at different time points, with 100 nM AHL induced for 24 h. (**e**) Quantitative analysis of the images in (**d**). Data represent the mean ± SD from 3 independent experiments. (**f**) Ultrasound images of GVs-miRFP680 MG1655 with a bacteria density of 0 (LB medium), 0.5 × 10^9^, 1.0 × 10^9^, and 2.0 × 10^9^/mL, with 100 nM AHL induced after 24 h. (**g**) Quantitative analysis of the images in (**f**). Data represent the mean ± SD from 3 independent experiments. **** *p* < 0.0001 vs. control. ** *p* < 0.01 vs. control.

**Figure 4 pharmaceutics-14-01804-f004:**
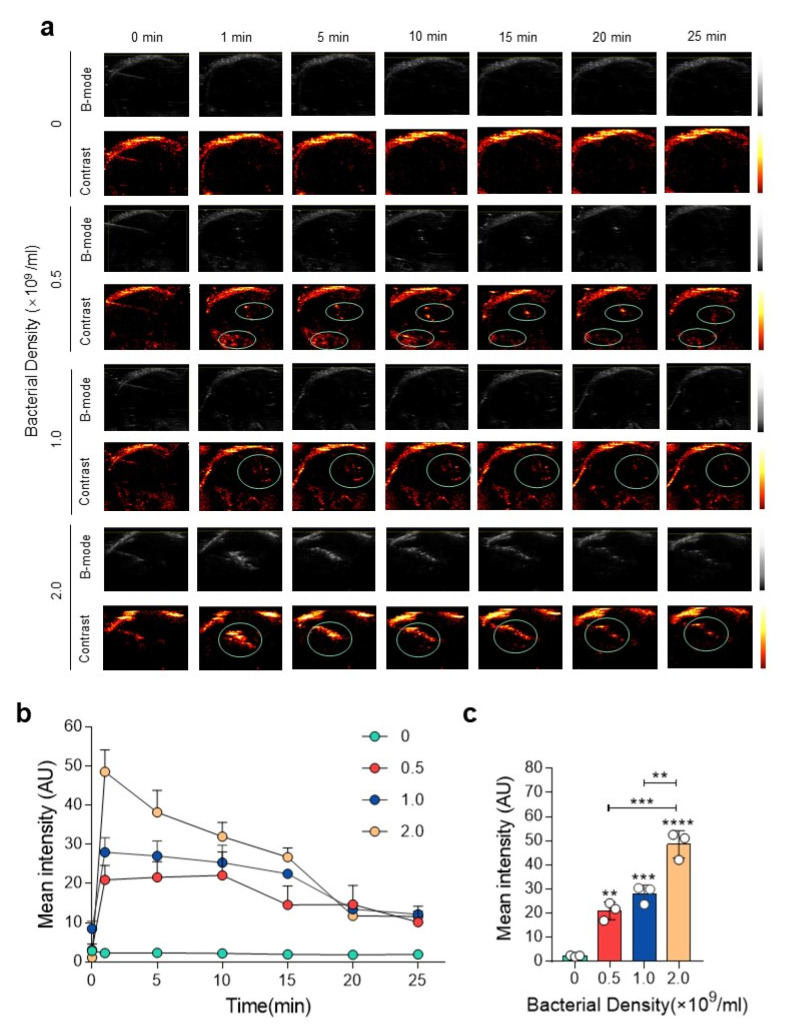
In vivo ultrasound imaging of GVs-miRFP680 MG1655. (**a**) In vivo acoustic imaging of the tumor was taken at different times after intratumoral injection of GVs-miRFP680 MG1655 with different bacterium concentrations (0, 0.5 × 10^9^, 1.0 × 10^9^, and 2.0 × 10^9^ CFU/mL). (**b**,**c**) Quantitative analysis of the acoustic imaging signal in (**a**). Data represent the mean ± SD from 3 independent experiments. **** *p* < 0.0001 vs. control. *** *p* < 0.001 vs. control. ** *p* < 0.01 vs. control.

**Figure 5 pharmaceutics-14-01804-f005:**
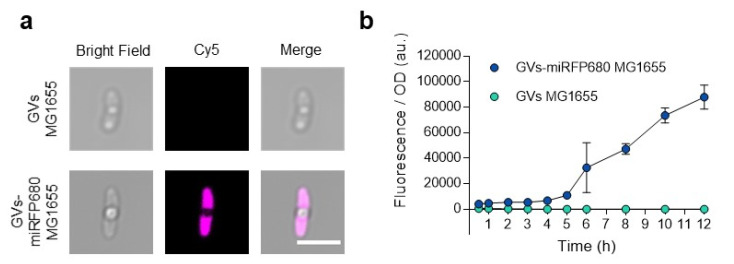
In vitro optical imaging of GVs-miRFP680 MG1655. (**a**) Confocal images of GVs MG1655(control) and GVs-miRFP680 MG1655 in the bright field, cy5 channel (emission: 640 nm, excitation: 663–738 nm), and merged channel, respectively. Scale bars = 5 μm. (**b**) Optical density (OD_600_)-normalized fluorescence as a function of culture time (ex = 661 nm, em = 680 nm). Data represent the mean ± SD from 3 independent experiments.

**Figure 6 pharmaceutics-14-01804-f006:**
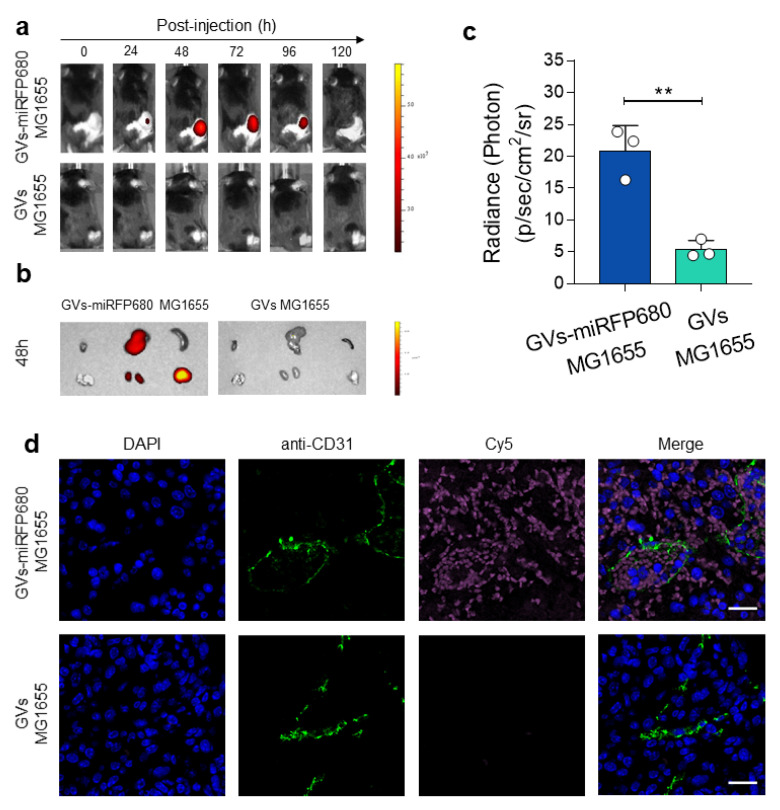
In vivo tracking capability of the tumor-homing characteristic of GVs-miRFP680 MG1655. (**a**) In vivo NIR fluorescent imaging of the tumor-bearing mice was taken at different times after intravenous injection of GVs-miRFP680 MG1655 or GVs MG1655(control). ex = 640 nm, em = 680 nm. (**b**) Ex vivo fluorescence imaging of organs and tumors taken from the tumor-bearing mice at 48 h post-injection. (**c**) Quantitative analysis for the fluorescence accumulated in the tumor in (**b**). ** *p* < 0.01 vs. control. (**d**) Confocal images of the tumor slices collected from the mice at 48 h post-injection of GVs-miRFP680 MG1655 or GVs MG1655 (control). The green and pink signals derived from the fluorescence of anti-CD31-stained blood vessels and the engineered GVs-miRFP680 MG1655 bacteria, respectively. Scale bars represent 20 μm.

## Data Availability

Not applicable.

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
