# Peer review of "Bimodal Imaging of Tumors via Genetically Engineered Escherichia coli"

_pharmaceutics, 2022, doi:10.3390/pharmaceutics14091804_

Round 1
Reviewer 1 Report
The authors demonstrated an engineered bacteria for tumor imaging. They acquired in vivo images of bacteria delivered to the tumor site with ultrasound and fluorescence imaging devices. The authors claim the results show the potential of their technique for tumor bacteriotherapy, but a comparison from previous research is missing in this manuscript. There are also some comments for improving the manuscript.
1. What is the main novelty of this study? Please clarify this.
2. It would be better to merge the Discussion and the Conclusion sections.
3. Please be specific in the title. The authors only used two biomedical imaging techniques: ultrasound and fluorescence imaging. The current title may mislead the readers.
4. Please do not use abbreviations in the Abstract
5. What is the rationale for using Escherichia Coli among the various bacterias?
6. In the Method section, the authors mentioned some portion of this study obtained from other research groups. Please mention this information in Acknowledge section. It is required to briefly explain the method although those materials from other labs.
7. Please also briefly summarize the TEM method although they followed previously reported protocol.
8. Is there any reference for tumor volume calculation?
9. Some abbreviations are used without explanation: ANOVA, CFU, PBS.
Please provide their full name although they are commonly used abbreviations.
10. There is no detailed explanation for all the figures. The author should guide the reader point by point on what the figures indicate.
11. What is the difference between B-mode and Contrast mode in ultrasound imaging? There is no explanation of how the authors acquired those images. Please provide a detailed experimental setup.
12. In fluorescence imaging, which wavelength was used for excitation? It would be good to provide the optical absorption spectrum of the engineered bacteria.
Overall, all the figures are too small to read.
Reviewer 2 Report
Zhang et al and their colleagues describe an approach for Bimodal imaging of tumors via genetically engineered Escherichia coli. It represents an enormous amount of work and brings about important new data and ideas. I believe that this manuscript can ultimately get acceptance on the condition of rigorous revision addressing the comments shown below:
1. kindly add graphical abstract to the main manuscript to draw the reader's attention.
2. Add the main results in the abstract.
3. Kindly add the ethical approval number in the animal models section.
4. Materials are missing in this manuscript, kindly update it, Add a separate section for cell lines, and explain the detailed protocol for it.
5. None of these tumors reached the maximal tumor size of 1,500 mm3? Why Justify it?
6. After the tumor volume reached an average size of 120 mm3, all experimental animals were randomly assigned to different groups in 126the following experiments. What kind of different group? kindly add group information in the mauscript.
7. Discussion: This part requires a thorough development. The authors should clarify the signalized doubts. They should demonstrate the advantages and disadvantages of the proposed system against the background of similar systems described earlier. The authors should also present their suggestions related to possible possibilities of the practical application of the described solution.
8. Please revisit the entire manuscript for minor grammar issues. The writing although good needs to be corrected for grammar and sentence construction. I also highly recommend the authors streamline their writing to keep the underlying conclusions precise and clear. The transitions between ideas seem disconnected. These would only help the reader get more from the review and improve its quality and appeal
Round 2
Reviewer 1 Report
The authors have carefully addressed the previous questions with the revised manuscript think now the manuscript is acceptable.
Reviewer 2 Report
Accepted